# Task-aware Distributed Source Coding under Dynamic Bandwidth

**Po-han Li**[1][*]   **Sravan Kumar Ankireddy**[1][*]   **Ruihan Zhao**[1]   **Hossein Nourkhiz Mahjoub**[2]

**Ehsan Moradi-Pari**[2]   **Ufuk Topcu**[1]   **Sandeep Chinchali**[1]   **Hyeji Kim**[1]

[1]The University of Texas at Austin [2]Honda Research Institute USA

## Abstract

Efficient compression of correlated data is essential to minimize communication overload in multi-sensor networks. In such networks, each sensor independently compresses the data and transmits them to a central node. A decoder at the central node decompresses and passes the data to a pre-trained machine learning-based task model to generate the final output. Due to limited communication bandwidth, it is important for the compressor to learn only the features that are relevant to the task. Additionally, the final performance depends heavily on the total available bandwidth. In practice, it is common to encounter varying availability in bandwidth. Since higher bandwidth results in better performance, it is essential for the compressor to dynamically take advantage of the maximum available bandwidth at any instant. In this work, we propose a novel distributed compression framework composed of independent encoders and a joint decoder, which we call neural distributed principal component analysis (NDPCA). NDPCA flexibly compresses data from multiple sources to any available bandwidth with a single model, reducing compute and storage overhead. NDPCA achieves this by learning low-rank task representations and efficiently distributing bandwidth among sensors, thus providing a graceful trade-off between performance and bandwidth. Experiments show that NDPCA improves the success rate of multi-view robotic arm manipulation by $9\%$ and the accuracy of object detection tasks on satellite imagery by $14\%$ compared to an autoencoder with uniform bandwidth allocation. [2]

## 1   Introduction

Efficient data compression plays a pivotal role in multi-sensor networks to minimize communication overload. Due to the limited communication bandwidth of such networks, it is often impractical to transmit all sensor data to a central server, and compression of the data is necessary. It is important for the sensors to compress the respective data independently, to avoid communication overload in the network. Information theory literature refers to this setting as distributed source coding [1], where the goal is to recover the original data with minimal distortion. In many cases, the data collected by the sensors is only processed by a downstream task model, *e.g.*, an object detection model, but not by humans, and hence the original distributed source coding goal of minimizing reconstruction error is no longer applicable. Instead, the goal should be to maximize the performance of the downstream task model. Additionally, in practice, data collected by multi-sensor networks is often correlated *e.g.* stereo cameras with overlapping fields of view. To improve communication efficiency, it is

---

[*]Equal contribution; order decided randomly. Correspondence to {pohanli, sravan.ankireddy}@utexas.edu.

[2] https://github.com/UTAustin-SwarmLab/Task-aware-Distributed-Source-Coding.

37th Conference on Neural Information Processing Systems (NeurIPS 2023).

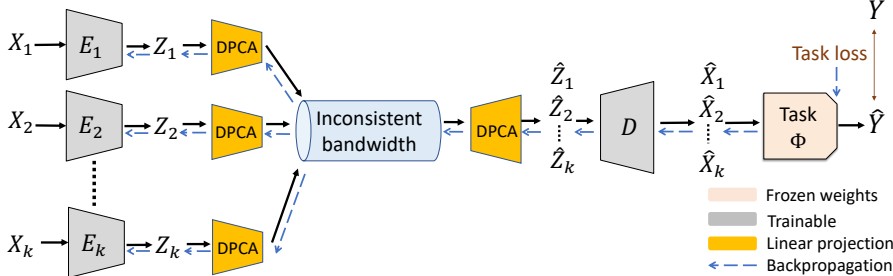

Figure 1: **Task-aware distributed source coding with NDPCA.** $X_1, \ldots, X_k$ are correlated data sources. Encoders $E_1, \ldots, E_k$ independently compress data to latent representations $Z_1, \ldots, Z_k$. Using linear matrices, the DPCA module projects the representations to any lower dimension at the encoder and projects them back to the original data space at the decoder, which allocates the bandwidth of sources based on the importance of the task $\Phi$. The goal is to find the optimal encoders and decoder that minimize the final task loss.

important for the compression framework to take advantage of the correlation and avoid transmission of redundant data. Combining both objectives, the final goal of the distributed compression framework is to learn relevant features that maximize the task performance, while avoiding the transmission of redundant features by exploiting the correlation between sources. Together, we refer to the distributed compression of task-relevant features as *task-aware distributed source coding*.

However, existing compression methods fail to combine the following three aspects: 1. Existing distributed compression methods perform poorly in the presence of a task model. Although neural networks have been shown to be capable of compressing stereo images [2, 3] and correlated images [4], existing methods focus on reconstructing image data, but not for downstream tasks. 2. Existing task-aware compression methods cannot take advantage of the correlation of sources. Previous works only consider compressing task-relevant features of single source [5–9], but not multiple correlated sources. 3. All existing methods for 1 & 2, especially those based on neural networks, only compress data to a fixed level of compression but not to multiple levels. Thus, they cannot operate in environments with different demands of compression levels and require a separate model trained for each compression level. Here, we note that we use the term bandwidth to indicate the information bottleneck in the dimension of transmitted data. Based on the choice of quantization, it is straightforward to convert the latent dimension to other popular metrics such as bits per pixel (bpp) in the case of image sources. Additionally, we consider the scenario of total bandwidth constraint for the uplink, which is typical for wireless networks [10].

We design neural distributed principal component analysis (NDPCA)–a distributed compression framework that can transmit task-relevant features at multiple compression levels. We consider the case where a task model at the central node requires data from all sources and the bandwidth in the network is not consistent over time, as shown in Fig. 1. In NDPCA, neural encoders $E_1, E_2, \ldots, E_K$ first independently compress correlated data $X_1, X_2, \ldots, X_K$ to latent representations $Z_1, Z_2, \ldots, Z_K$. A proposed module called distributed principal component analysis (DPCA) further compresses these representations to any lower dimension according to the current bandwidth and decompresses the data at the central node. Finally, a neural decoder at the central node decodes the representations $\hat{Z}_1, \hat{Z}_2, \ldots, \hat{Z}_k$ to $\hat{X}_1, \hat{X}_2, \ldots, \hat{X}_K$ and feeds them into a task. Task-aware compression aims to minimize task loss, defined as the difference in task outputs with and without compression, such as the difference in object detection results. Due to the significant training cost involved, we avoid training the task model, which is usually a large pre-trained neural network.

To highlight our proposed method, NDPCA learns task-relevant representations with a single model at multiple compression levels. It includes a neural autoencoder to generate uncorrelated task-relevant representations in a fixed dimension. It is desirable to learn uncorrelated representations to prevent the transmission of redundant information. It also includes a module for linear projection, DPCA, to allocate the available bandwidth among sources based on the importance of the task, by observing the respective principal components, and then further compressing the representations to any desired dimension. By harmoniously combining the neural autoencoder and the linear DPCA module, NDPCA generates representations that are more compressible in limited bandwidths, providing a graceful trade-off between performance and bandwidth.

**Contributions:** Our contributions are three-fold: First, we formulate a task-aware distributed source coding problem that optimizes the compression for a given task instead of reconstructing the sources (Sec. 2). Second, we provide a theoretical justification for the framework by analyzing the case of a linear compressor and a linear task (Sec. 3). Finally, we propose a task-aware distributed source coding framework, NDPCA, that learns a single model for different levels of compression to handle any type of source and task(Sec. 4). We validate NDPCA with tasks of CIFAR-10 image denoising, multi-view robotic arm manipulation, and object detection of satellite imagery (Sec. 5). NDPCA results in a 1.2dB increase in PSNR, a $9\%$ increase in success rate, and a $14\%$ increase in accuracy compared to an autoencoder with uniform bandwidth allocation, for the respective experiments mentioned above.

## 2   Problem Formulation

We now define the problem statement more formally. Consider a set of $K$ correlated sources. Let $x_i \in \mathbb{R}^{n_i}$ denote the sample from source $i$ where $i \in \{1, 2, \ldots, K\}$. Samples from each source $i$ are compressed independently by encoder $E_i$ to a latent representation $z_i \in \mathbb{R}^{m_i}$ such that $\sum_{i=1}^{K} m_i = m$, where $m$ is the total bandwidth available. A joint decoder $D$ receives the representations $\{z_1, z_2, \ldots, z_k\}$ and reconstructs the sources $\{\hat{x}_1, \hat{x}_2, \ldots, \hat{x}_k\}$. In the setting without a task, the goal is to find a set of encoders and a decoder to recover the inputs $\{x_1, x_2, \ldots, x_k\}$ with minimal loss:

$$\operatorname*{argmin}_{E_1, E_2, \ldots, E_k, D} \sum_{i=1}^{K} \mathcal{L}_{\text{rec}}(x_i, \hat{x}_i) \quad (\textit{Task-agnostic distributed source coding}), \tag{1}$$

where $\mathcal{L}_{\text{rec}}$ is the reconstruction loss, *e.g.*, the mean-squared error loss.

In the presence of a task $\Phi$, it takes the reconstructed inputs to compute the final output $\Phi(\hat{x}_1, \hat{x}_2, \ldots, \hat{x}_k)$. The goal is to find a set of encoders and a decoder such that the task loss $\mathcal{L}_{\text{task}}$ is minimized, where $\Phi(x_1, x_2, \ldots, x_k)$ is the task output computed without compression. We refer to this problem as *task-aware* distributed source coding, which is the main focus of this paper:

$$\operatorname*{argmin}_{E_1, E_2, \ldots, E_k, D} \mathcal{L}_{\text{task}}(\Phi(x_1, x_2, \ldots, x_k), \Phi(D(E_1(x_1), E_2(x_2), \ldots, E_k(x_k))))$$
$$(\textit{Task-aware distributed source coding}), \tag{2}$$

where $\mathcal{L}_{\text{task}}$ is the task loss, *e.g.*, the difference of bounding boxes when the task is object detection.

**Bandwidth allocation:** In the previous formulations, we assume that the output dimensions of encoders are known a priori. However, the dimensions are related to the compression quality of each encoder, which is also a design factor. That is, given the total available bandwidth $m$, we first need to obtain the optimal $m_i$ for each source $i$, then, we can design the optimal encoders and decoder accordingly. Finding the optimal set of bandwidths for a given task is a long-standing open problem in information theory [11], even for the simple task of a modulo-two sum of two binary sources [12]. Also, existing works [4, 13, 14] largely assume a fixed latent dimension for sources and train different models for different total available bandwidth $m$, which is, of course, suboptimal. In this paper, our framework provides heuristics to the underlying key challenge of optimally allocating available bandwidth, *i.e.*, deciding $m_i$, while adapting to different total bandwidths $m$ with a single model.

## 3   Theoretical Analysis

We start with a motivating example of task-aware distributed source coding under the constraint of linear encoders, a decoder, and a linear task. We first solve the linear setting using our proposed method, distributed principal component analysis (DPCA). We then describe how DPCA compresses data to different bandwidths and analyze the performance of DPCA. In this way, we gain insights into combining DPCA with neural autoencoders in later Sec. 4.

**DPCA Formulation:** We consider a linear task for two sources, defined by the task matrix $\Phi \in \mathbb{R}^{p \times (n_1 + n_2)}$, where the sources $x_1 \in \mathbb{R}^{n_1}$ and $x_2 \in \mathbb{R}^{n_2}$ are of dimensions $n_1$ and $n_2$, respectively, and the task output is given by $y = \Phi x \in \mathbb{R}^p$, where $x = [x_1^\top, x_2^\top]^\top$. Without loss of generality, we assume the sources to be zero-mean. Now, we have $N$ observations of two sources $X_1 \in \mathbb{R}^{n_1 \times N}$ and $X_2 \in \mathbb{R}^{n_2 \times N}$ and their corresponding task outputs $Y = \Phi(X) \in \mathbb{R}^{p \times N}$, where $X = [X_1^\top X_2^\top]^\top$.

We aim to design the optimal linear encoding matrices (encoders) $E_1 \in \mathbb{R}^{m_1 \times n_1}$, $E_2 \in \mathbb{R}^{m_2 \times n_2}$, and the decoding matrix (decoder) $D \in \mathbb{R}^{(n_1+n_2) \times (m_1+m_2)}$ that minimizes the task loss defined as the Frobenius norm of $\Phi(X) - \Phi(\hat{X})$, where $\hat{X}$ is the reconstructed $X$. We only consider the non-trivial case where the total bandwidth is less than the task dimension, $m = m_1 + m_2 < p$, i.e., the encoders cannot directly calculate the task output locally and transmit it to the decoder. For now, we assume that $m_1$ and $m_2$ are given, and we discuss the optimal allocation later in this section.

Letting $Z_1 = E_1 X_1 \in \mathbb{R}^{m_1 \times N}$ and $Z_2 = E_2 X_2 \in \mathbb{R}^{m_2 \times N}$ denote the encoded representations and $M = \Phi D$ denote the product of the task and decoder matrices, we solve the optimization problem:

$$E_1^*, E_2^*, M^* = \operatorname*{argmin}_{E_1, E_2, M} \quad \|Y - MZ\|_2^2 \tag{3a}$$

$$\text{s.t.} \quad Z = \begin{bmatrix} Z_1 \\ Z_2 \end{bmatrix} = \begin{bmatrix} E_1 X_1 \\ E_2 \tilde{X}_2 \end{bmatrix}, \tag{3b}$$

$$ZZ^\top = \mathbb{I}_m, \tag{3c}$$

$$\hat{Y} = \Phi DZ = MZ, \quad Y = \Phi \begin{bmatrix} X_1 \\ X_2 \end{bmatrix}. \tag{3d}$$

Note that solving $M$ is identical to solving the decoder $D$ since we can always convert $M$ to $D$ by the generalized inverse of task $\Phi$. The encoders $E_1$ and $E_2$ project the data to representations $Z_1$ and $Z_2$ in (3b). We constrain the representations to be orthonormal vectors in (3c) as in the normalization in principal component analysis (PCA) for the compression of a single source [15]. This constraint lets us decouple the problem into subproblems later in (5). Finally, in (3d), the decoder $D$ decodes $Z_1$ and $Z_2$ to $\hat{X}_1$ and $\hat{X}_2$ and passes the reconstructed data to task $\Phi$.

**Solution:** We now solve the optimization problem in (3). For any given $E_1, E_2$ (thus, a given $Z$), we can optimally obtain $M^* = YZ^\top(ZZ^\top)^{-1} = YZ^\top$ by linear regression. Now, we are left to find the optimal encoders $E_1, E_2$. First, a preprocessing step removes the correlation part of $X_1$ from $X_2$ by subtracting the least-square estimator $\hat{X}_2(X_1)$:

$$\tilde{X}_2 = X_2 - \hat{X}_2(X_1) = X_2 - X_2 X_1^\top (X_1 X_1^\top)^{-1} X_1. \tag{4}$$

The orthogonality principle of least-square estimators [16, p.386] ensures that $X_1 \tilde{X}_2^\top = \mathbf{0}_{n_1 \times n_2}$. We decouple the objective in (3a) with respect to $E_1, E_2$ by the orthogonality principle and (3c):

$$\min_{E1, E2} \|Y - M^* Z\|_2^2 = \|Y\|_2^2 - \max_{E_1, E_2} \|M^*\|_2^2 = \|Y\|_2^2 - \max_{E_1} \|Y_1 X_1^\top E_1^\top\|_2^2 - \max_{E_2} \|Y_2 \tilde{X}_2^\top E_2^\top\|_2^2, \tag{5}$$

where $Y = \Phi X = [\Phi_1 \Phi_2] \left[ X_1^\top X_2^\top \right]^\top = Y_1 + Y_2$. We then have two subproblems from (3):

$$E_1^* = \operatorname*{argmax}_{E_1} \quad \|\Phi_1 X_1 X_1^\top E_1^\top\|_2^2 \qquad \qquad E_2^* = \operatorname*{argmax}_{E_2} \quad \|\Phi_2 \tilde{X}_2 \tilde{X}_2^\top E_2^\top\|_2^2$$

$$\text{s.t.} \quad E_1 X_1 X_1^\top E_1^\top = \mathbb{I}_{m_1}, \tag{6} \qquad \qquad \text{s.t.} \quad E_2 \tilde{X}_2 \tilde{X}_2^\top E_2^\top = \mathbb{I}_{m_2}. \tag{7}$$

The two subproblems are the canonical correlation analysis [17], which can be solved by whitening $E_1 X_1$, $E_2 \tilde{X}_2$ and singular value decomposition (see [17] for details).

**Dynamic bandwidth:** So far, we solved the case for fixed bandwidths $m_1$ and $m_2$. We now describe ways to determine the optimal bandwidth allocation given a current total bandwidth $m$. To do so, DPCA solves (6) and (7) with $m_1 = n_1$ and $m_2 = n_2$ and obtains $E_1^*$, $E_2^*$ and all pairs of canonical directions and correlations. Canonical directions and correlations can be analogized to a more general case of singular vectors and values. Similar to PCA, the sums of squares of canonical correlations are the optimal values of (6) and (7), so DPCA sorts all the canonical correlations in descending order and chooses the first $m$ pairs of canonical correlations and directions. These canonical correlations determine the optimal encoders $E_1^*, E_2^*$ and decoder $D^*$, which indirectly solves $m_1$ and $m_2$. Intuitively, the canonical correlations indicate the importance of a direction to the task, and we prioritize the transmission of directions by importance. For simplicity, we only consider the case of 2 sources. DPCA can easily compress more sources simply by constraining all $Z$s to be independent and thus decoupling the original problem (3) to more subproblems.

**Performance analysis of DPCA:** When DPCA compresses new data matrices with encoder $E_1^*$ and $E_2^*$, the preprocessing step is invalid as the encoders cannot communicate with each other. So for

DPCA to perform optimally while skipping the step, the two data matrices need to be uncorrelated, namely, $\hat{X}_2(X_1) = 0$, because in such case the preprocessing step removes nothing from the data sources. Given that correlated sources lead to suboptimality of DPCA, we characterize the performance between the joint compression, PCA, and the distributed compression, DPCA, under the same bandwidth in Lemma 3.1 with the simplest case of reconstruction, namely, $\Phi = \mathbb{I}_p$. In this setting, the canonical correlation analysis is relaxed to the singular value decomposition, which is later used for NDPCA in Sec. 4.

**Lemma 3.1** (Bounds of DPCA Reconstruction). *Given a zero-mean data matrix and its covariance,*

$$X = \begin{bmatrix} X_1 \\ X_2 \end{bmatrix} \in \mathbb{R}^{(n_1+n_2) \times N}, XX^\top = \underbrace{\begin{bmatrix} \mathrm{Cov}_{11} & \mathbf{0} \\ \mathbf{0} & \mathrm{Cov}_{22} \end{bmatrix}}_{X_{\mathrm{diag}}} + \underbrace{\begin{bmatrix} \mathbf{0} & \mathrm{Cov}_{12} \\ \mathrm{Cov}_{21} & \mathbf{0} \end{bmatrix}}_{\Delta X},$$

*assume that $\Delta X$ is relatively smaller than $XX^\top$, and $XX^\top$ is positive definite with distinct eigenvalues. For PCA's encoding and decoding matrices $E_{\mathrm{PCA}}, D_{\mathrm{PCA}}$ and DPCA's encoding and decoding matrices $E_{\mathrm{DPCA}}, D_{\mathrm{DPCA}}$, the difference of the reconstruction losses is bounded by*

$$0 \leq \|X - D_{\mathrm{DPCA}} \, E_{\mathrm{DPCA}}(X)\|_2^2 - \|X - D_{\mathrm{PCA}} E_{\mathrm{PCA}}(X)\|_2^2 = - \sum_{i=m+1}^{n_1+n_2} \lambda_i e_i^\top \Delta X e_i.$$

*where $\lambda_i$ and $e_i$ are the $i$-th largest eigenvalue and eigenvector of $XX^\top$, $\mathrm{Tr}$ is the trace function, and $m$ is the dimension of the compression bottleneck.*

The proof of Lemma 3.1 is in Appendix A.1. Note that $\Delta X$ is the correlation of sources, so as $\|\Delta X\|_F$ gets smaller, the difference of PCA and DPCA is closer to 0. That is, as the covariance decreases, DPCA performs more closely to PCA, which is the optimal joint compression.

To summarize, uncorrelated data matrices $X_1, \ldots, X_K$ are desired for DPCA. If so, DPCA optimally decides the bandwidths of all sources based on the canonical correlations, representing their importance for the task. One application of DPCA is that encoders can use the remaining unselected canonical directions to improve compression when the available bandwidth is higher later.

## 4 Neural Distributed Principal Component Analysis

The theoretical analysis in the previous section indicates that DPCA has two drawbacks: it only compresses data optimally if sources are uncorrelated, and it only works for linear tasks. However, DPCA dynamically allocates bandwidth to sources based on their importance. On the other hand, neural autoencoders are shown to be powerful tools for compressing data to a fixed dimension but cannot dynamically allocate bandwidth. This contrast motivates us to harmoniously combine a neural autoencoder to generate representations and then pass them through DPCA to compress and find the bandwidth allocation. We refer to the combination of a neural autoencoder and DPCA as neural distributed principal component analysis (NDPCA). With a single neural autoencoder and a matrix at each encoder and decoder, NDPCA adapts to any available bandwidth and flexibly allocates bandwidth to sources according to their importance to the task.

**Outline:** NDPCA has two encoding stages, as shown in Fig. 1: First, the neural encoder at each $k$-th source encodes data $X_k$ to a fixed-dimensional representation $Z_k$ for $k \in [K]$. Then the DPCA linear encoder adapts the dimension of $Z_k$ via linear projection according to the available bandwidth and the correlation among the sources as per (6). Similarly, the decoding of NDPCA is also performed in two stages. First, the DPCA linear decoder reconstructs the $K$ fixed-dimensional representations $\hat{Z}_1, \hat{Z}_2, \ldots, \hat{Z}_K$, based on which the joint neural decoder generates the estimate of data $\hat{X}_1, \hat{X}_2 \ldots, \hat{X}_K$. These estimates are then passed to the neural task model $\Phi$ to obtain the final task output $\hat{Y}$. Note that since we have a non-linear task model here, we envision that the neural encoders generate non-linear embedding of the sources, while the DPCA mainly adapts the dimension appropriately as needed; the role of the DPCA here is to reliability reconstruct the embedding $\hat{Z}$s, which corresponds to the case described in Lemma 3.1 with the task matrix $\Phi$ as identity.

**Training procedure:** During the training of NDPCA, the weights of the task are always frozen because it is usually a large-scale pre-trained model that is expensive to re-train. We aim to learn the

$K$ neural encoders and the joint neural decoder which minimize the loss function:

$$\mathcal{L}_{\text{tot}} = \lambda_{\text{task}} \underbrace{\|\hat{Y} - Y\|_F^2}_{\text{task loss}} + \lambda_{\text{rec}} \underbrace{\left( \|\hat{X}_1 - X_1\|_F^2 + \|\hat{X}_2 - X_2\|_F^2 + \dots \|\hat{X}_K - X_K\|_F^2 \right)}_{\text{reconstruction loss}}. \quad (8)$$

In the task-aware setting when $\lambda_{\text{rec}} = 0$, the neural autoencoder fully restores task-relevant features, which is the main focus of this paper. When $\lambda_{\text{task}} = 0$, the neural autoencoder learns to reconstruct the data $X$, which is the task-agnostic setting later compared in Sec. 5.

We now discuss how to encourage NDPCA to work well under *various available bandwidths* with DPCA during the training phase. We begin by making observations on the desired property of the neural embeddings arising from the limitations of the DPCA: (1) uncorrelatedness: Lemma 3.1 shows that DPCA is more efficient when the correlation among the intermediate representations is less. (2) linear compressibility: we encourage the neural autoencoder to generate low-rank representations, which can be compressed by only a few singular vectors, making them more bandwidth efficient.

We tried to explicitly encourage the desired properties with additional terms in (8), but they all adversely affect the task performance. To obtain uncorrelated representations, we tried penalizing the cosine similarity between the representations. We also tried similar losses that penalize correlation, as per [18–21], but none improves the task performance. We observed that the autoencoder automatically learns representations with small correlation, and any explicit imposition of complete uncorrelatedness is too strong. For linear compressibility, we tried penalizing the convex low-rank approximation–the nuclear norm–of the representations, as per [22, 23]. However, we observe a similar trend in the final task performance as the network tends to minimize the nuclear norm while harming the task performance. For the comparison of the resulting performance, see Appendix F.1.

In this regard, we propose a novel linear compression module that allows us to adapt to DPCA during training rather than using additional terms in the loss. We introduce a *random-dimension* DPCA projection module to improve performance in lower bandwidths. It projects representations $Z$ to a low dimension randomly chosen, simulating projections in various available bandwidths during inference. It can be interpreted as a differentiable singular value decomposition with a random dimension, described in Alg. 1. For encoding, it first normalizes the representations and performs singular value decomposition on all sources. Then, it sorts the vectors by the singular values and randomly selects the number of vectors to use for projection. For decoding, it decodes with the selected singular vectors again and denormalizes the data. Note that during training, we only run Alg. 1 on a batch. This module helps to improve the overall performance over a range of bandwidths, and we show the ablation study of this module in Appendix F.2.

**Inference:** With the training data, the DPCA projection module first saves the mean of representations $Z$ and the encoder and decoder matrices in the maximum bandwidth. It only needs to save for the maximum bandwidth because its rows and columns are already sorted by the singular values, which represent the importance of each corresponding vector. During inference, when the current bandwidth is $m$, it chooses the top $m$ rows and columns of the saved encoders and decoder matrices to encode and decode representations. No retraining is needed for different bandwidths. Only the storage of a neural autoencoder and a linear matrix at each encoder and decoder is needed.

**Robust task model:** We pre-train the task model with randomly cropped and augmented images to make the model less sensitive to noise in the input image space, namely, the model has a smaller Lipschitz constant. This augmentation trick is based on [8]. A robust task model has a smaller Lipschitz constant, so it is less sensitive to the input noise injected by decompression when we concatenate it with the neural autoencoder. For a detailed analysis of the performance bounds between robust task and task-aware autoencoders, see Appendix A.2.

## 5 Experiments

We consider three different tasks to test our framework: ($a$) the denoising of CIFAR-10 images [24], ($b$) multi-view robotic arm manipulation [25], which we refer to as the *locate and lift* task, and ($c$) object detection on satellite imagery [26]. Across all the experiments, we assume that there are two data sources, referred to as *views*, each containing partial information relevant to the task. We present our results based on the testing set and refer to our proposed method, task-aware NDPCA, as NDPCA for simplicity. NDPCA includes a single autoencoder with a large dimension of representations

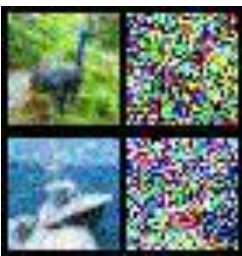

(a) CIFAR-10: view 1 is less corrupted and thus contains more information about the original images.

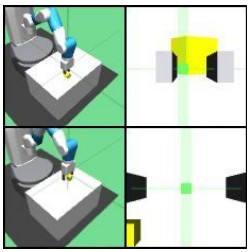

(b) Locate and lift: Side-view (column 1) faintly captures the absolute position of objects, in contrast to the arm-view (column 2).

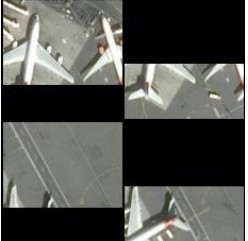

(c) Airbus detection: view 1 and view 2 observe different parts of the complete view with overlap.

Figure 2: **Datasets:** (column 1) view 1. (column 2) view 2. In all experiments, both views are correlated, but one view is more important than the other as it contains more information relevant to the task.

$Z \in \mathbb{R}^{2*m_{\max}}$. It then compresses representations and allocates bandwidth via DPCA, as discussed previously. We show that NDPCA can bridge the performance gap between distributed autoencoders and joint autoencoders, defined below, to allocate bandwidth and avoid transmitting task-irrelevant features. We also provide experiments of NDPCA with more than 2 data sources in Appendix C to demonstrate NDPCA's capability in more complicated settings.

**Baselines:** We compare NDPCA against three major baselines. First is the task-aware joint autoencoder (JAE), where a single pair of encoder and decoder compresses both views. JAE is considered an upper bound of NDPCA since it can leverage the correlation between both views while avoiding encoding redundant information. Next is the task-aware vanilla distributed autoencoder (DAE), where two encoders independently encode the corresponding views to equal bandwidths and a joint decoder decodes the data. DAE is considered a lower bound of NDPCA since both encoders utilize the same bandwidth regardless of the importance of the views for the task, while

---

**Algorithm 1** Projection into a random low dimension using DPCA

1: **Input:** A size $b$ batch of latent representations $Z_i \in \mathbb{R}^{b \times m_i}$ from each source $i$, min and max bandwidth $m_{\min}, m_{\max}$
2: **Output:** Compressed representation $Z_i^m$ of each source, reconstructed representation $\hat{Z}$ for all sources
3: **function** ENCODE($Z_i, m_{\min}, m_{\max}$)
4:     **for** each source $i$ **do**
5:         $\bar{Z}_i \leftarrow Z_i - \text{Mean}(Z_i)$       ▷ Normalize representations
6:         $s_i, V_i, H_i \leftarrow \text{SVD}(\bar{Z}_i)$     ▷ Singular value decomposition
7:     **end for**
8:     $s, V \leftarrow \text{Cat}(s_i), \text{Cat}(V_i)$ ▷ Concatenate singular values and vectors
9:     $m \leftarrow \text{Rand}(m_{\min}, m_{\max})$ ▷ Randomly choose projection dimension
10:    $s^m, V^m \leftarrow \arg\max([s, V], m)$    ▷ Select the top $m$ values of $s$
11:    **for** each source $i$ **do**
12:        $V_i^m \leftarrow \{V | V \in V^m, V \in V_i\}$ ▷ Select $m$ vectors from sources
13:        $Z_i^m = \bar{Z}_i \times V_i^m$       ▷ Project $Z_i$ to lower dimensions
14:    **end for**
15:    **return** $Z_{\text{low}} \leftarrow \text{Cat}(Z_i^m)$   ▷ Return Compressed representation
16: **end function**
17: **function** DECODE($Z_i^m$)
18:    **for** each source $i$ **do**
19:        $\hat{\bar{Z}}_i \leftarrow Z_i^m \times \text{Cat}(V_i^m)^\top$   ▷ Decompressed representation
20:        $\hat{Z}i \leftarrow \hat{\bar{Z}}_i + \text{Mean}(Z_i)$    ▷ Denormalize representations
21:    **end for**
22:    **return** $\hat{Z} \leftarrow \text{Cat}(\hat{Z}i)$    ▷ Return reconstructed representations
23: **end function**

---

NDPCA allocates bandwidths in a task-aware manner. Last is the task-agnostic NDPCA which differs from NDPCA in the training loss of reconstructing the original views. Due to the novelty of the problem formulation, we cannot make a fair comparison with any of the existing approaches. For instance, [2–4, 14] focus purely on distributed compression of images for reconstruction and human perception, whereas [6, 7] focus on task-oriented compression but are limited for a single source. Additionally, none of the previous works consider datasets of unequal importance, again making any performance comparison unfair. Hence we focus mainly on an ablation study style of comparison of NDPCA, clearly highlighting and validating the advantages of our approach.

**CIFAR-10 denoising:** We first consider a simple task of denoising CIFAR-10 images using two noisy observations of the same image, shown in Fig. 2(a). We use CIFAR-10 as a toy example to clearly highlight the advantage of NDPCA in the presence of sources with unequal importance to the task. Due to the simplistic nature of the classification task, which only requires 4 bits (digit 0-9) as

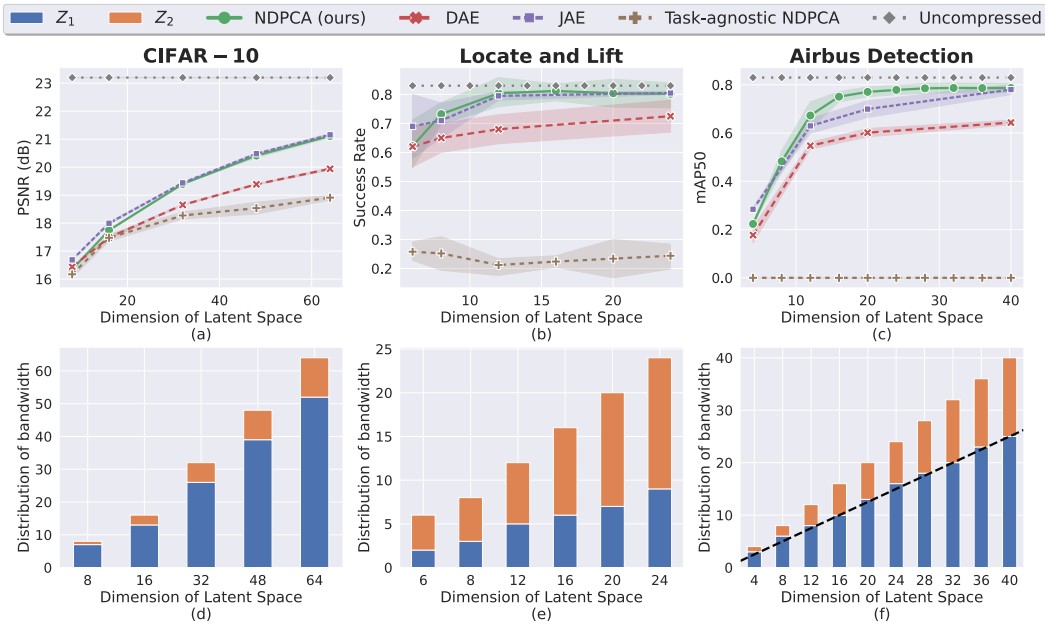

Figure 3: **Top:** Performance Comparison for 3 different tasks. Our method achieves equal or higher performance than other methods. **Bottom:** Distribution of total available bandwidth (latent space) among the two views for NDPCA (ours). The unequal allocation highlights the difference in the importance of the views for a given task.

the information bottleneck, we choose denoising as our "task", making it more suitable to showcase the performance across a range of available bandwidth. Here, the importance of each observation, or view, for the task is simply the noise level. For view 1, we consider an image corrupted with additive white Gaussian noise (AWGN) with a variance of $0.1^2$. And view 2 is highly corrupted by AWGN with a variance of 1. All the images were normalized to $[0, 1]$ before adding the noise. We compressed the noisy observations and passed the reconstructed images through a pre-trained denoising network. We then computed the final peak signal-to-noise ratio (PSNR) with respect to the clean image. Since the noise levels of both views are unequal, the importance of the task is unequal as well. The optimal bandwidth allocation should not be equal, thus showing the advantage of NDPCA. Although view 1 contains more information, not all bandwidth should be allocated to view 1. This problem is called the CEO problem [27, 28]. In fact, even if one view is highly corrupted, we should still leverage that view and never allocate 0 bandwidth to it. We discuss why it is the case in Appendix B.

**Locate and lift:** For the manipulation task, we consider a scenario in which a simulated 6 degrees-of-freedom robotic arm controlled by a reinforcement learning agent inputs two camera views to locate and lift a yellow brick. We call the view from the robotic arm "arm-view" and the one recording the whole desk "side-view", as shown in Fig. 2 (b). The two views are complementary to completing the task, details discussed in Appendix F.3. We trained the agent in a supervised-learning manner. We collected a dataset of observation and action pairs [29] and trained an agent from the dataset. Then, we defined task loss as the $L_2$ norm of actions from images with and without compression and trained NDPCA to minimize the task loss through the agent. Literature calls this training method "behavior cloning" [30] as it learns from demonstrations. Behavior cloning causes a drop in performance, but this paper only focuses on the performance degradation caused by compression, so we treat the behavior cloning agent with uncompressed views as the upper bound of our method.

**Airbus detection:** This task considers using satellite imagery to locate Airbuses. Satellites observe overlapping images of an airport and transmit data to Earth through limited bandwidth, as shown in Fig. 2 (c). We crop all images in the dataset into smaller pieces ($224 \times 224$ pixels). The two data sources are the upper 160 pixels (source 1) and the lower 104 pixels of the image (source 2) with 40 pixels overlapped. Our object detection model follows the paper "You Only Look Once" (Yolo) [31]. The task loss here is the difference between object detection loss with and without compression.

**Results:** Our key results are: (1) Task-aware NDPCA outperforms task-agnostic NDPCA, and (2) bandwidth allocation should be related to the importance of the task. Across all experiments, shown in Fig. 3(a)-(c), we see that task-aware NDPCA performs much better than task-agnostic NDPCA

and DAE, which equally allocates bandwidths. We see from Fig. 3 that task-aware NDPCA provides a graceful performance degradation with respect to available bandwidth, with no additional training or storage of multiple models. On the other hand, DAE and JAE require retraining for every level of compression, so every sample point in the plot is a different model.

Fig. 3(a) shows the results of denoising CIFAR-10 with NPDCA trained at $(m_{\min}, m_{\max}) = (8, 64)$. Although view 1 is more important than view 2, DAE can only equally allocates bandwidth to both sources. NDPCA compresses the data and flexibly allocates bandwidths, as shown in 3(d), where we can see that $Z_1$ has more bandwidth than $Z_2$. NDPCA results in 1.2 dB gain in PSNR compared to DAE when $m = 64$.

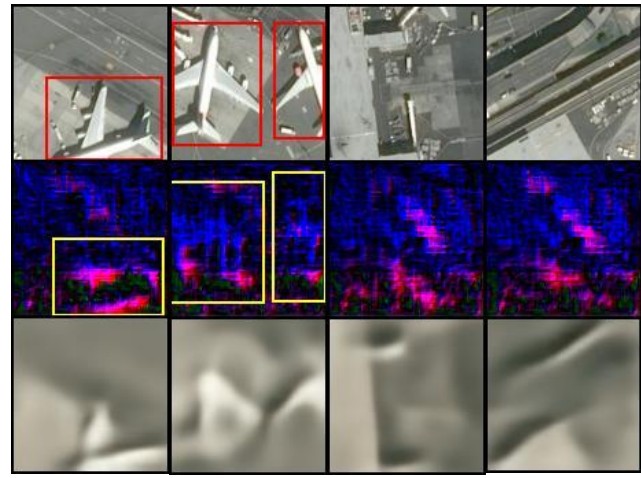

Figure 4: **Task-aware v.s. task-agnostic:** Ground-truth bounding boxes are red (row 1), while detected boxes of task-aware are yellow (row 2). Nothing is detected in the task-agnostic setting (row 3). Task-agnostic images are perceptible to human eyes, while task-aware images capture task-relevant features, thus imperceptible to human eyes.

Fig. 3(b) shows the results of the locate and lift task with NPDCA trained at $(m_{\min}, m_{\max}) = (8, 48)$. We set the length of an episode as 50 time steps and measure the success rate in 100 episodes. We show the upper bound, a behavior cloning agent without compression, in gray dotted lines. The arm view is more important as it captures the precise location of the brick, and as expected, NDPCA allocates more bandwidth to the arm-view ($Z_2$), as seen in Fig. 3(e). We see that NDPCA has a $9\%$ higher success rate compared to DAE when $m = 24$.

Fig. 3(c) shows the results of the Airbus detection with NPDCA trained at $(m_{\min}, m_{\max}) = (8, 40)$. We measured the mean average precision (mAP) with $40\%$ confidence score and $50\%$ intersection over the union as the thresholds. We show the uncompressed upper bound in gray dotted lines. NDPCA results in up to $14\%$ gain in mAP50 compared to DAE. In Fig. 3(f), we plotted the ratio of the areas of both views, while equally splitting the overlapping part, in a dashed black line. Surprisingly, NDPCA's empirical allocation of bandwidth is highly aligned with the theoretical ratio, supporting that it captures the importance of the task and allocates bandwidth according to it.

**Comparison of NDPCA with JAE**: JAE uses the information from both views simultaneously to capture the best joint embedding for the task. In an ideal scenario, JAE will be the upper bound for the performance and hence easily performs better than DAE across all the experiments. Interestingly, in Fig. 3(b) and (c), we see that NDPCA outperforms not only DAE but also JAE as well. We attribute it to the better representations present in higher-dimension latent space. It turns out that learning a high-dimensional representation and then projecting to a lower dimension space, like NDPCA, is more efficient compared to directly learning a low-dimensional representation, like JAE. This projection from higher dimensional to lower dimensional is similar to pruning large neural networks to identify effective sparse sub-networks. [32, 33]. We also note that Low-Rank Adaptation (LoRA) [34] technique for large language models can be thought of as a similar approach.

**Task-aware v.s. task-agnostic:** We plotted the reconstructed images of task-aware ($\lambda_{\rm rec} = 0$) and task-agnostic ($\lambda_{\rm task} = 0$) NDPCA in Fig. 4. Task-aware images are imperceptible to human eyes since they restore features of a non-linear task model, aligning with the results in [8, Fig. 4]. For discussion of non-zero $\lambda_{\rm task}$ and $\lambda_{\rm rec}$, we refer readers to Appendix E.5.

**Limitations:** In general, autoencoders are poor at generalizing to out-of-distribution data and the drawback translates to NDPCA as well. When the testing set is noticeably different from the training set, the performance of NDPCA can get noticeably lower. Additionally, during training, DPCA performs the singular value decomposition in the training set. The decomposition operation can become ill-conditioned and unstable if the batch size is too small. An alternative approach could be a

parametric low-rank decomposition such as LoRA [34] or using adapter networks [35], although the complexity increases and the compatibility with DPCA remains to be explored.

# 6    Related Work

**Information theoretic perspective:** Slepian and Wolf *et al.* are the first to obtain the minimum bandwidth of distributed sources to perfectly reconstruct data [36]. However, they use exponentially complex compressors while assuming that the joint distribution of sources is known, which is impractical. In the presence of a task, finding the rate region of two binary sources has remained an open problem, even for modulo-two sum tasks [12]. In terms of imperfectly reconstructing data with neural autoencoders, previous works consider compression of the original data to a fixed dimension [13, 37], while our work focuses on compressing data to any bandwidth with a task model.

**Task-aware compression:** Real-world data, such as images or audio, are ubiquitous and high-dimensional, while downstream tasks that input the data only utilize certain features for the output. Task-aware compression aims to compress data while maximizing the performance of a downstream task. Previous works analyze linear task [5], image compression [6–8, 38], future prediction [9], and data privacy [39, 40], while ours compresses distributed sources under limited bandwidth.

**Neural autoencoder:** Previous works show the ability of neural autoencoders to generate meaningful and uncorrelated representations. Instead of adding additional loss terms during training like [18–21, 41], we use a random projection module to help a neural autoencoder learn uncorrelated and linear-compressible representations. Other works focus on designing new neural architectures for multi-view image compression [4, 14], while ours focuses on the framework to compress data to different compression levels. We choose autoencoders instead of variational autoencoders [42, 43] because we focus on the compression of fixed representations rather than generative tasks from latent distributions. Also, autoencoders are more compatible with DPCA than variational autoencoders.

# 7    Conclusion and Future Work

We proposed a theoretically grounded linear distributed compressor, DPCA, and analyzed its performance compared to the optimal joint compressor. Then, we designed a distributed compression framework called NDPCA by combining a neural autoencoder and DPCA to allocate bandwidth according to their importance to the task. Experiments on CIFAR-10 denoising, locate and lift, and Airbus detection showed that NDPCA near-optimally outperforms task-agnostic or equal-bandwidth compression schemes. Moreover, NDPCA requires only one model and does not need to be retrained for different compression levels, which makes it suitable for settings with dynamic bandwidths.

Avenues for future research include settings where the information flow is not unidirectional but bidirectional, such that the encoders and the decoder can communicate to compress data better. Discovering representations in a more complex space using kernel PCA instead of linear PCA and exploration of more complex non-linear correlations are also left as interesting future work. Another interesting direction to expand the work would be analyzing the robustness of the representations, both in the latent space with respect to corruption such as additive white Gaussian noise (AWGN) as well as with respect to the downstream task model. The current framework learns task-relevant features that are tied to the task model but the performance is expected to drop significantly when the task model is updated or changed. Hence, it is desirable to incorporate robust and transferable properties into the features learned.

## Acknowledgement

This work was supported in part by the National Science Foundation 2133481, NASA 80NSSC21M0071, ARO Award W911NF2310062, ONR Award N00014-21-1-2379, NSF Award CNS-2008824, and Honda Research Institute through 6G@UT center within the Wireless Networking and Communications Group (WNCG) at the University of Texas at Austin. Any opinions, findings, and conclusions or recommendations expressed in this material are those of the authors and do not necessarily reflect the views of the National Science Foundation.

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
