# Appendix

## A Proofs of Lemmas

### A.1 Bounds of DPCA

**Lemma** (Bounds of DPCA Reconstruction). *Given a zero-mean data matrix and its covariance,*

$$
X = \begin{bmatrix} X_1 \\ X_2 \end{bmatrix} \in \mathbb{R}^{(n_1+n_2) \times N}, XX^\top = \underbrace{\begin{bmatrix} \mathrm{Cov}_{11} & \mathbf{0} \\ \mathbf{0} & \mathrm{Cov}_{22} \end{bmatrix}}_{X_{\mathrm{diag}}} + \underbrace{\begin{bmatrix} \mathbf{0} & \mathrm{Cov}_{12} \\ \mathrm{Cov}_{21} & \mathbf{0} \end{bmatrix}}_{\Delta X},
$$

*assume that $\Delta X$ is relatively smaller than $XX^\top$, and $XX^\top$ is positive definite with distinct eigenvalues. For PCA's encoding and decoding matrices $E_{\mathrm{PCA}}, D_{\mathrm{PCA}}$ and DPCA's encoding and decoding matrices $E_{\mathrm{DPCA}}, D_{\mathrm{DPCA}}$, the difference of the reconstruction losses is bounded by*

$$
0 \le \|X - D_{\mathrm{DPCA}} \, E_{\mathrm{DPCA}}(X)\|_2^2 - \|X - D_{\mathrm{PCA}} E_{\mathrm{PCA}}(X)\|_2^2 = - \sum_{i=m+1}^{n_1+n_2} \lambda_i e_i^\top \Delta X e_i.
$$

*where $\lambda_i$ and $e_i$ are the $i$-th largest eigenvalue and eigenvector of $XX^\top$, $\mathrm{Tr}$ is the trace function, and $m$ is the dimension of the compression bottleneck.*

*Proof.* The lower bound is intuitive. We know that DPCA cannot outperform PCA since distributed coding cannot outperform joint coding and PCA is the optimal linear encoding. The reconstruction loss of PCA is always not greater than the loss of DPCA, thus the lower bound is $0$. Now consider the upper bound:

$$
\begin{aligned}
&\|X - D_{\mathrm{DPCA}} E_{\mathrm{DPCA}} X\|_2^2 - \|X - D_{\mathrm{PCA}} E_{\mathrm{PCA}} X\|_2^2 \\
&= \mathrm{Tr}(XX^\top + D_{\mathrm{DPCA}} E_{\mathrm{DPCA}} X (D_{\mathrm{DPCA}} E_{\mathrm{DPCA}} X)^\top - 2 D_{\mathrm{DPCA}} E_{\mathrm{DPCA}} XX^\top) \\
&\quad - \sum_{i=m+1}^{n_1+n_2} \lambda_i(XX^\top) \\
&= \mathrm{Tr}(X_{\mathrm{diag}} + \Delta X + D_{\mathrm{DPCA}} E_{\mathrm{DPCA}} X (D_{\mathrm{DPCA}} E_{\mathrm{DPCA}} X)^\top - 2 D_{\mathrm{DPCA}} E_{\mathrm{DPCA}} XX^\top) \\
&\quad - \sum_{i=m+1}^{n_1+n_2} \lambda_i(XX^\top) \\
&= \mathrm{Tr}(\Delta X + E_{\mathrm{DPCA}}^\top D_{\mathrm{DPCA}}^\top D_{\mathrm{DPCA}} E_{\mathrm{DPCA}} \Delta X - 2 D_{\mathrm{DPCA}} E_{\mathrm{DPCA}} \Delta X) \\
&\quad + \sum_{i=m+1}^{n_1+n_2} \lambda_i(X_{\mathrm{diag}}) - \lambda_i(XX^\top) \\
&= \sum_{i=m+1}^{n_1+n_2} \lambda_i(X_{\mathrm{diag}}) - \lambda_i(XX^\top).
\end{aligned}
$$

Finally, we use the matrix perturbation theory [44] to calculate the first-order approximation of the effect of $\Delta X$ on the singular values of $X_{\mathrm{diag}}$. The perturbation theory assumes that the perturbation

$\Delta X$ is relatively small compared to $X_{\text{diag}}$. Then, we know:

$$\|X - D_{\text{DPCA}} E_{\text{DPCA}} X\|_2^2 - \|X - D_{\text{PCA}} E_{\text{PCA}} X\|_2^2 = \sum_{i=m+1}^{n_1+n_2} \lambda_i(X_{\text{diag}}) - \lambda_i(XX^\top)$$

$$\leq \sum_{i=m+1}^{n_1+n_2} \lambda_i - \lambda_i - \lambda_i e_i^\top \Delta X e_i$$

$$= -\sum_{i=m+1}^{n_1+n_2} \lambda_i e_i^\top \Delta X e_i.$$

$\square$

Note that the encoding and decoding matrices of DPCA look like:

$$D_{\text{DPCA}} = \begin{bmatrix} D_1 & \mathbf{0} \\ \mathbf{0} & D_1 \end{bmatrix}, E_{\text{DPCA}} = \begin{bmatrix} E_1 & \mathbf{0} \\ \mathbf{0} & E_2, \end{bmatrix}$$

where $E_1, E_2, D_1, D_2$ are matrices obtained from each source with DPCA.

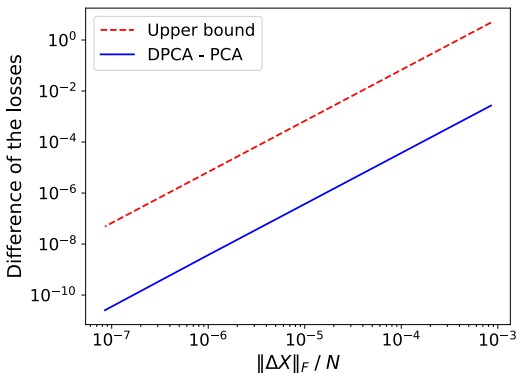

Figure 5: **Bound from Lemma 3.1**: The obtained upper bound is always larger than the difference of losses of DPCA and PCA.

We examine the correctness of our bound with random data matrices in Fig. 5. We can see that the gap between DPCA and PCA decreases as the Frobenius norm of $\Delta X$ decreases. The upper bound also has the same trend, while it is always larger than the exact value. Note that in Fig. 5, all axes are in log scale.

## A.2   Why Robust Task?

We now characterize the effect of using task-aware compression and a pre-trained, robust task. We assume that the robust task performs similarly to the original, non-robust task. We also know that the robust task has a lower Lipschitz constant than the non-robust one [45, 46]. We denote the robust task model by $\Phi^*$ and the non-robust task model by $\Phi$. We define task-aware autoencoder as

$$D_{\text{awa}}, E_{\text{awa}} = \arg\min_{D,E} \quad \|\Phi^*(x) - \Phi^* \circ D \circ E(x)\|_2^2$$

$$\text{subject to} \quad E(x) \in \mathbb{R}^\Phi,$$

and task-agnostic autoencoder as

$$D_{\text{agn}}, E_{\text{agn}} = \arg\min_{D,E} \quad \|x - D \circ E(x)\|_2^2$$

$$\text{subject to} \quad E(x) \in \mathbb{R}^\Phi,$$

where $\circ$ denotes function composition. For simplicity, we further define

$$\hat{x}_{\text{awa}} = D_{\text{awa}} \circ E_{\text{awa}}(x), \quad \hat{x}_{\text{agn}} = D_{\text{agn}} \circ E_{\text{agn}}(x).$$

Then, we prove the following lemma:

**Lemma A.1** (Why task-aware compression and a robust task). *Assume robust task model $\Phi^*$ and non-robust task $\Phi$ only differ in:*

$$\forall x, \quad \|\Phi^*(x) - \Phi(x)\| \leq \epsilon. \tag{9}$$

*That is, the robust task and the normal task have a bounded performance gap. Assume that $\Phi^*$ is a Lipschitz function with constant $L^*$, and $\Phi$ is a bi-Lipschitz function with constant $L$. Namely,*

$$\|\Phi^*(x) - \Phi^*(\tilde{x})\|_2 \leq L^* \|x - \tilde{x}\|_2, \tag{10}$$

*and*

$$\frac{1}{L}\|x - \tilde{x}\|_2 \leq \|\Phi(x) - \Phi(\tilde{x})\|_2 \leq L\|x - \tilde{x}\|_2. \tag{11}$$

*We show that the task losses of task-aware, robust models and task-agnostic, non-robust models are bounded by*

$$\|\Phi^*(x) - \Phi^*(\hat{x}_{\text{awa}})\|_2 - L^*\|x - \hat{x}_{\text{awa}}\|_2 + \frac{1}{L}\|x - \hat{x}_{\text{agn}}\|_2$$
$$\leq \|\Phi(x) - \Phi(\hat{x}_{\text{agn}})\|_2 \tag{12}$$
$$\leq \|\Phi^*(x) - \Phi^*(\hat{x}_{\text{awa}})\|_2 + 2\epsilon + L^*\|\hat{x}_{\text{awa}} - \hat{x}_{\text{agn}}\|_2.$$

*Proof.* We consider the difference between the two task losses. By the triangle inequality,

$$\|\Phi(x) - \Phi(\hat{x}_{\text{agn}})\|_2 - \|\Phi^*(x) - \Phi^*(\hat{x}_{\text{awa}})\|_2$$
$$\leq \|\Phi(x) - \Phi(\hat{x}_{\text{agn}}) - \Phi^*(x) + \Phi^*(\hat{x}_{\text{awa}})\|_2$$
$$= \|\Phi(x) - \Phi^*(x) + \Phi^*(\hat{x}_{\text{awa}}) - \Phi(\hat{x}_{\text{agn}})\|_2$$
$$\leq \|\Phi(x) - \Phi^*(x)\|_2 + \|\Phi^*(\hat{x}_{\text{awa}}) - \Phi(\hat{x}_{\text{agn}})\|_2 \tag{13}$$
$$\leq \epsilon + \|\Phi^*(\hat{x}_{\text{awa}}) - \Phi^*(\hat{x}_{\text{agn}}) + \Phi^*(\hat{x}_{\text{agn}}) - \Phi(\hat{x}_{\text{agn}})\|_2$$
$$\leq \epsilon + \|\Phi^*(\hat{x}_{\text{awa}}) - \Phi^*(\hat{x}_{\text{agn}})\|_2 + \|\Phi^*(\hat{x}_{\text{agn}}) - \Phi(\hat{x}_{\text{agn}})\|_2$$
$$\leq 2\epsilon + L^*\|\hat{x}_{\text{awa}} - \hat{x}_{\text{agn}}\|_2.$$

On the other hand, subtracting (10) and (11), we get

$$\|\Phi^*(x) - \Phi^*(\hat{x}_{\text{awa}})\|_2 - \|\Phi(x) - \Phi(\hat{x}_{\text{agn}})\|_2$$
$$\leq L^*\|x - \hat{x}_{\text{awa}}\|_2 - \frac{1}{L}\|x - \hat{x}_{\text{agn}}\|_2. \tag{14}$$

Finally, combining (13) and (14), we get

$$\|\Phi^*(x) - \Phi^*(\hat{x}_{\text{awa}})\|_2 - L^*\|x - \hat{x}_{\text{awa}}\|_2 + \frac{1}{L}\|x - \hat{x}_{\text{agn}}\|_2$$
$$\leq \|\Phi(x) - \Phi(\hat{x}_{\text{agn}})\|_2$$
$$\leq \|\Phi^*(x) - \Phi^*(\hat{x}_{\text{awa}})\|_2 + 2\epsilon + L^*\|\hat{x}_{\text{awa}} - \hat{x}_{\text{agn}}\|_2.$$

$\square$

Lemma A.1 characterizes how close the task losses of task-aware robust models and task-agnostic non-robust models are. The reason that robust task models are preferable to non-robust models is that robust task models have smaller Lipschitz constants. In other words, when noise caused by communication or reconstruction perturbs the input of the models, the output is less sensitive, so the output of the perturbed task is closer to the original output.

With regard to task-aware autoencoders, it is obvious that they are preferable to task-agnostic ones since the former minimizes task losses. Task-agnostic autoencoders aim to reconstruct the full image, but most pixels in an image are not related to the task, so task-agnostic models are more bandwidth inefficient than task-aware models. Of course, when one has sufficient bandwidth to transmit a whole image perfectly, task-agnostic models will perform equally to task-aware models. In this case, $\|x - \hat{x}_{\text{awa}}\|_2 = \|x - \hat{x}_{\text{agn}}\|_2 = 0$ in (12).

## B The Gaussian CEO Problem

The Gaussian CEO problem [27, 28] refers to the problem of distributed inference from noisy observations. The objective is to reconstruct the source from noisy observations rather than the noisy observations themselves, which motivated our first experiment of CIFAR-10 denoising. In the original setting, a White Gaussian source $X$ of variance $P$ is observed through two independent Gaussian broadcast channels $Y_j = X + Z_j$ for $i = 1, 2$ where $Z_1 \sim \mathcal{N}(0, N_1)$ and $Z_2 \sim \mathcal{N}(0, N_2)$. The observations $Y_1$ and $Y_2$ are separately encoded with the aim of estimating $X$ such that the mean square error distortion between the estimate $\hat{X}$ and $X$ is $D$.

The rate-distortion region $R_{\text{CEO}}(D)$ for the quadratic Gaussian CEO problem is the set of rate pairs $(R_1, R_2)$ that satisfy

$$R_1 \geq r_1 + \frac{1}{2} \log D - \frac{1}{2} \log \left( \frac{1}{P} + \frac{1 - e^{-2r_2}}{N_2} \right), \tag{15}$$

$$R_2 \geq r_2 + \frac{1}{2} \log D - \frac{1}{2} \log \left( \frac{1}{P} + \frac{1 - e^{-2r_1}}{N_1} \right), \tag{16}$$

for some $r_1, r_2 \geq 0$ such that

$$D \geq \left( \frac{1}{P} + \frac{1 - e^{-2r_1}}{N_1} + \frac{1 - e^{-2r_2}}{N_2} \right)^{-1}. \tag{17}$$

Considering the CIFAR-10 denoising experiment, we have $P = 0.3125$, and for a target distortion of PSNR 20 dB, we have $D = 0.01$. For the sake of analysis, we assume the CIFAR-10 source to be Gaussian and find the lower bounds on rates $R_1$ and $R_2$. We begin by solving for the auxiliary variables that satisfy (17). Then, in the region of feasible auxiliary rates, we look for the pair of $(r_1, r_2)$ that minimize the sum lower bound on $R_1 + R_2$. Solving this for $N_1 = 0.01$ and $N_2 = 1$, we get $R_1 \geq 3.44$ and $R_2 \geq 0.002$. Similarly, for $N_1 = 0.01$ and $N_2 = 0.1$, we get $R_1 \geq 2.45$ and $R_2 \geq 0.41$. Under the assumption of Gaussian sources, this clearly demonstrates that the rates for both sources are non-zero. Also, the rate allocated to a source is inversely proportional to the noise. Therefore, $R_1 > R_2$ when source 1 is less noisy, implying that higher bandwidth is allocated to source 1 since it contains more *information* and is more *important*.

## C NDPCA with 4 sources:

To showcase the capability of NDPCA under more than 2 sources, we examine it on the most complicated dataset amomng the 3–Airbus detection. Views 1 and 2 have resolutions of $(160 \times 224)$ pixels, whereas views 3 and 4 have $(288 \times 224)$ pixels. Same as the previous experiments, we intentionally set the views to different sizes so that the importance to the task is unequal, resulting in different bandwidth allocatation among the sources in Fig. 6.

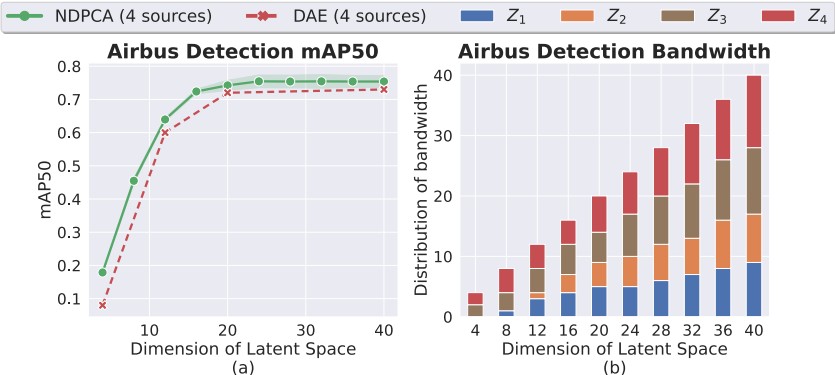

Figure 6: **NDPCA with 4 data sources:** (a) The performance of NPCA compared to DAE with 4 data sources. NDPCA is better than DAE, which aligns with our result with 2 data sources. (b) Distribution of total available bandwidth (latent space) among the 4 views for NDPCA. The difference in resolution emphasizes the distinct importance of each view in object detection, therefore $Z_3$ and $Z_4$ have greater dimensions than $Z_1$ and $Z_2$.

# D Details of the Datasets

## D.1 CIFAR-10 denoising:

We started with the standard CIFAR-10 dataset and normalized the images to $[0, 1]$. Two different views are created by adding different levels of Gaussian noise, $\mathcal{N}(0, 0.1^2)$ and $\mathcal{N}(0, 1)$. The pretrained task model is created by training a denoising autoencoder that takes both views, concatenates them along the channel dimension, and produces a clean image. The autoencoders need to learn features that are important for this task model.

## D.2 Locate and lift:

We collected $20,000$ pairs of actions and the corresponding images of both views for our training set. The actions are $4$ dimensional, controlling the $x, y, z$ coordinate movements and the gripper of the robotic arm. We randomly cropped the images from $128 \times 128$ to $112 \times 112$ pixels to make our autoencoder more robust. The expert agent is pre-trained by the same data augmentation as well.

## D.3 Airbus detection:

We first cropped all original images of $2560 \times 2560$ pixels (Fig. 7) into $224 \times 224$ pixels with 28 pixels overlapping between each cropped image. We then eliminated the bounding boxes that are less than $30\%$ left after cropping.

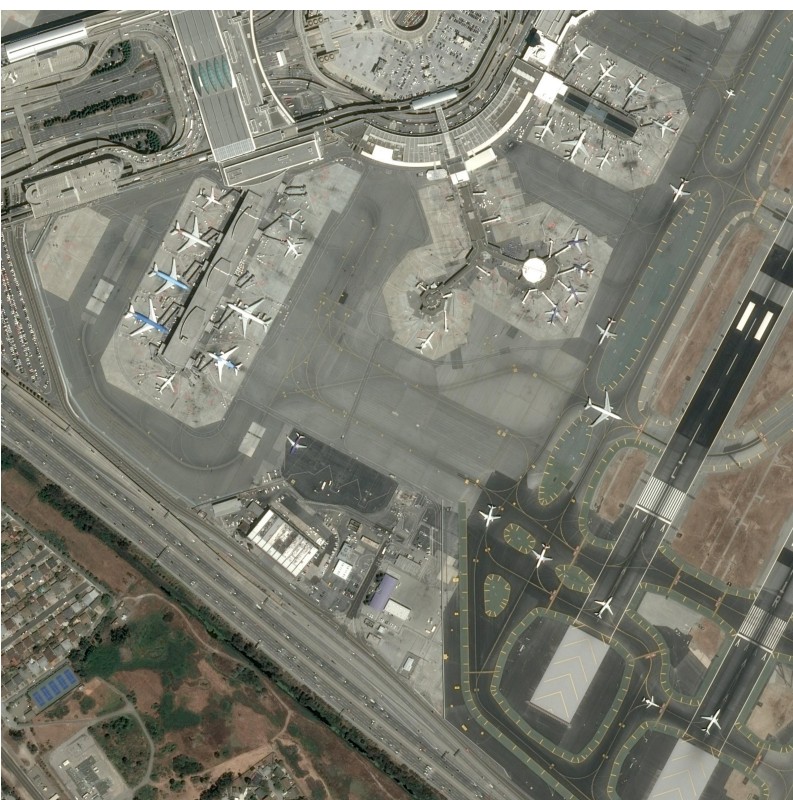

Figure 7: **Original image of airbus detection**. The original images are $2560 \times 2560$ pixels, and we cropped them into smaller pieces in $224 \times 224$.

# E   Implementation Details

## E.1   CIFAR-10 denoising:

For the CIFAR-10 dataset, we used the standard CIFAR-10 dataset and applied different levels of AWGN noise to create two correlated datasets. We used the CIFAR-10 experiments as a proof of concept to try different architectures and loss functions and other techniques to finalize our framework. We choose $\lambda_{\text{task}} = 1$ for the task-aware setting and $\lambda_{\text{rec}} = 1$ for the task-agnostic setting. We run 4 random seeds on NDPCA and all baselines to evaluate the performance.

## E.2   Locate and lift:

For the locate and lift experiment, we trained our autoencoder with the same random cropping setting as in Sec. D, which cropped the images from $128 \times 128$ to $112 \times 112$ pixels. During testing, we randomly initialized the location of the brick and center-cropped the images from $128 \times 128$ to $112 \times 112$ pixels. We scaled all images to 0 to 1 and ran 5 random seeds on NDPCA and all baselines to evaluate the performance. For the task-aware setting, $\lambda_{\text{task}} = 500$, and $\lambda_{\text{rec}} = 1000$ for the task-agnostic. setting

## E.3   Airbus detection:

For the Airbus detection task, we used the original Yolo paper for our object detection model together with the detection loss [31]. Our experiments with the latest state-of-the-art Yolo v8 model [47] showed that there is no big difference in the Airbus detection dataset in terms of run time and accuracy. Since the size of the original dataset is not enough to train an object detection model, we used the data augmentation proposed in Yolo v8, mosaic, to increase the size of the dataset. Mosaic randomly crops 4 images and merges them to generate a new image. We used random resized crop, blur, median blur, and CLAHE enhancement during training, each with probability 0.05 by functions in the Albumentations package [48]. We increased the size of the Airbus dataset from 5904 to 21808 with mosaic and trained the Yolo detection model. Finally, we trained our autoencoder with the same dataset, but downsample the images to $112 \times 112$ pixels so that the autoencoder is faster to train. For the task-aware setting, $\lambda_{\text{task}} = 0.1$, and $\lambda_{\text{rec}} = 0.5$ for the task-agnostic setting. We run 2 random seeds on NDPCA and all baselines to evaluate the performance.

## E.4   Neural Autoencoder Architecture and Hyperparameters

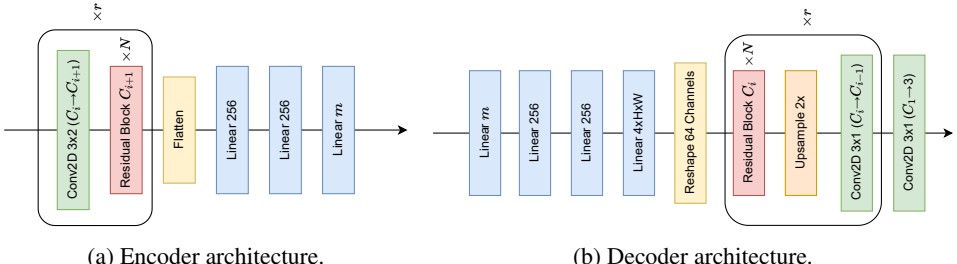

(a) Encoder architecture.     (b) Decoder architecture.

Figure 8: **ResNet Autoencoer:** The encoder processes inputs through $r$ convolution layers and $r \times N$ residual blocks, followed by 3 fully connected layers with ReLU activation. The decoder processes latent representations in the reverse order from the encoder with $2\times$ upsamplings.

We used the ResNet encoder shown in Fig. 8a and the decoder in Fig. 8b for all experiments. We used different numbers of filters and numbers of residual blocks for our experiments, shown as $C$ and $r$. We denote $m$ as the number of latent dimensions. The numbers of filters are $C_1 = 32$, $C_2 = 64$, $C_3 = 128$, $C_1 = 8$, $C_2 = 16$, $C_3 = 32$, $C_4 = 64$, and $C_1 = 16$, $C_2 = 32$, $C_3 = 64$, $C_4 = 128$, and the numbers of residual blocks are $r = 0$, $r = 1$, $r = 1$ for CIFAR-10 denoising, locate and lift, and Airbus detection. For CIFAR-10 denoising, we use the Adam optimizer with a learning rate of 0.0002, and for the other two experiments, we use the Adam optimizer with a learning rate of 0.0001. For the sake of training speed, when training DAE and JAE, we first trained a large network with

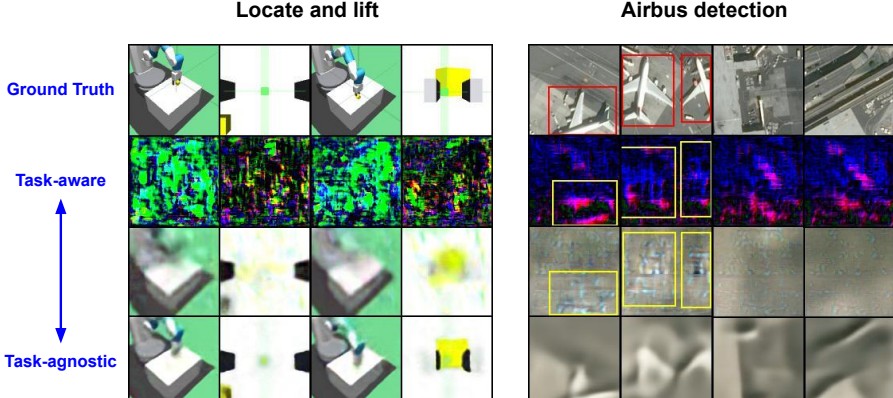

Figure 9: **Weighted task-loss:** Weighted task-aware images faintly reconstruct the original images while restoring task-relevant features with high-frequency noise. In Airbus detection, location of Airbuses is captured with shiny high-frequency pixels in row 3.

$m_{\max}$ with each random seed. Then, we fixed the network parameters and trained concatenate 3 fully connected layers on each encoder and decoder network to compress and decompress the data to smaller $m$.

## E.5 Balancing Task-aware and Task-agnostic Loss

NPDCA has a loss function consisting of 2 terms, as shown in (8):

$$\mathcal{L}_{\text{tot}} = \lambda_{\text{task}} \underbrace{\|\hat{Y} - Y\|_F^2}_{\text{task loss}} + \lambda_{\text{rec}} \underbrace{\left( \|\hat{X}_1 - X_1\|_F^2 + \|\hat{X}_2 - X_2\|_F^2 + \ldots \|\hat{X}_K - X_K\|_F^2 \right)}_{\text{reconstruction loss}}.$$

(8 revisited)

Previously, we tested two extreme cases of (8): task-aware when $\lambda_{\text{task}} > 0, \lambda_{\text{rec}} = 0$, and task-agnostic when $\lambda_{\text{task}} = 0, \lambda_{\text{rec}} > 0$. Of course, one can use different weighted sums of the 2 terms in (8), which we call weighted task-aware. We show the resulting reconstructed image in Fig. 9, whose weights are a mixture of half of the two other methods. Weighted task-aware images have both blurry reconstructions of the original images and task-relevant features. Unsurprisingly, the task loss and the reconstructed loss of weighted task-aware images are between pure task-aware and task-agnostic, that is, we can use the weights in the loss function to trade off compressing human perception features against task-relevant features. Interestingly, we can see that the task-aware images look similar to the images without Airbuses (last 2 columns), and when there are Airbuses, the task-aware images look different. It means that the features of no Airbuses are pretty much the same in the latent space, thus resulting in similar images in pixel space. Hence we can conclude that task-aware features are not random noise, they are meaningful features only to the task model but not to our eyes.

## E.6 Storage and Training Complexity

| Model | CIFAR-10 | | Locate and lift | | Airbus detection | |
|---|---|---|---|---|---|---|
| | Storage (MB) | Train (hr) | Storage (MB) | Train (hr) | Storage (MB) | Train (hr) |
| NDPCA | 8.3 | 0.25 | 16.4 | 5.0 | 33.0 | 13.0 |
| DAE | $5 \times 8.4$ | $5 \times 0.21$ | $4 \times 16.3$ | $4 \times 5.0$ | $4 \times 22.5$ | $4 \times 11.5$ |
| JAE | $5 \times 10.2$ | $5 \times 0.22$ | $4 \times 11.4$ | $4 \times 3.5$ | $4 \times 32.9$ | $4 \times 10.5$ |

Table 1: **Storage and training complexity:** NDPCA has slightly more storage and training overload than other models for a single bandwidth but can operate across different bandwidths. We multiply the number of bandwidths tested in Fig. 3 to the storage size and training time of DAE and JAE as they require different models for different compression levels.

One key feature of NDPCA is that it only needs one model to operate in different bandwidths. Therefore, we only need to train and store one model at the edge devices and the central node. We compare the complexity of storage and training in Table 1. Although NDPCA has a larger storage size and longer training time than other models, it can operate across different bandwidths. According to Table 1, if all models operate in more than 1 bandwidths, NDPCA saves more storage and training overload because other models have more than 50% of NPDCA's overload. For CIFAR-10 denoising, we tested the training time on an RTX 4090, and for the locate and lift and Airbus detection experiments, we tested the training time on an NVIDIA RTX A5000.

## F  Ablation Study

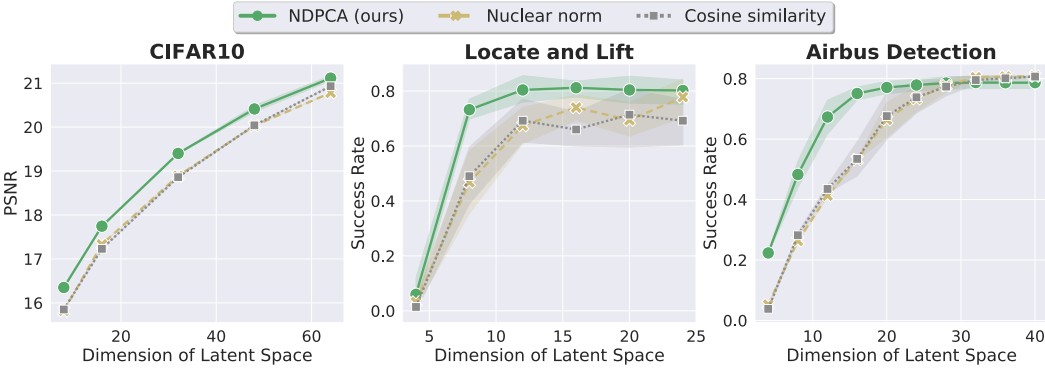

Figure 10: **Ablation study of the nuclear norm and cosine similarity:** Adding the nuclear norm or cosine similarity to the loss function does not improve the performance of the model when compressing latent representations to lower dimensions.

### F.1  Cosine similarity and nuclear norm

In Fig. 10, we show that adding nuclear norm or cosine similarity in the training loss (8) does not help the model perform when we use DPCA to project latent representations into lower dimensions. We compared our proposed NDPCA with the DPCA module against NDPCA without the DPCA module but with the penalization of the nuclear norm and cosine similarity added. The weights of all the additional terms are 0.1. From Fig. 10, we conclude that the DPCA module can increase the performance better than the other two.

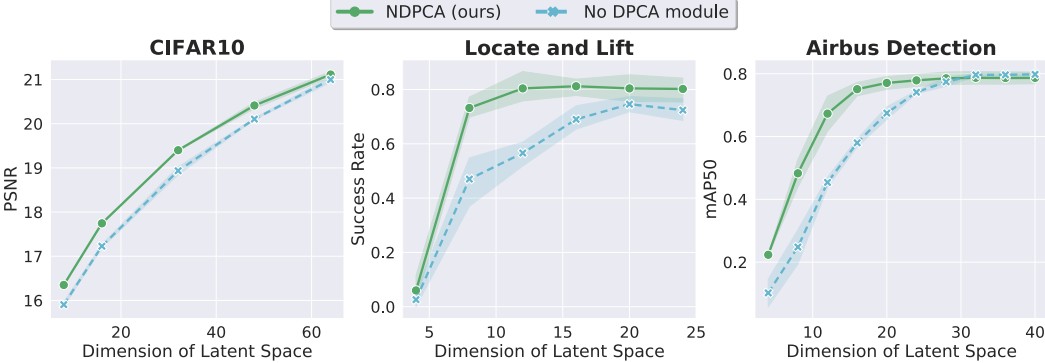

Figure 11: **Ablation study of DPCA module:** The proposed DPCA module effectively increases the performance in lower bandwidths, while achieving the same performance at larger bandwidths.

### F.2  DPCA module

In Fig. 11, we show that the proposed DPCA module can help the neural autoencoder learn linear compressible representations, as described in Sec. 4. We see that with the DPCA module, NDPCA

can increase the performance in lower bandwidths, while saturating at the performance close to the model without the module. We conclude that with the DPCA module, NDPCA learns to generate low-rank representations, so the performance is better in lower bandwidths. However, when the bandwidth is higher, the bandwidth can almost fully restore the representations, so the two methods perform similarly.

### F.3 Single view performance of locate and lift

In the locate and lift experiments, the reinforcement learning agent leverages information from both views as input to manipulate. Here, we detail why the 2 views are complementary to accomplish the task. The success rate of an agent is $76\%$ with only the arm-view and $45\%$ with the side-view. When combining both, the success rate is $83\%$. The reason why the views are complementary is that the side-view provides global information on the position of the arm and the brick, but sometimes the brick is hidden behind the arm. The arm-view captures detailed information from a narrow view of the desk. Once the arm-view captures the brick, it is straightforward to move toward it and lift it. The arm view is more important because with only the arm-view, the agent can randomly explore the brick, but with only the side-view, the brick might be vague to see and thus harder to lift. Of course, with both views, the robotic arm can easily move toward the vague position of the brick and use arm-view to lift it.