# OpenReview forum: "Task-aware Distributed Source Coding under Dynamic Bandwidth"
_NeurIPS.cc/2023/Conference — NeurIPS 2023 poster_

### Official Review · Reviewer_84yk · 2023-07-05

**Soundness:** 2 fair
**Presentation:** 2 fair
**Contribution:** 2 fair
**Rating:** 5
**Confidence:** 3

**Summary:**

This paper proposed a task-aware distributed source coding framework called NDPCA (Neural Distributed Principal Component Analysis). This framework aimed to solve the problem of efficient compression of correlated data in multi-sensor networks. In section 2 and 3, the authors provided a formulation of the problem and solved the problem in a linear setting with their proposed method DPCA. They also described ways to determine bandwidth allocation and analyzed the bound of DPCA reconstruction loss in practical conditions. In section 4, the authors proposed NDPCA to generalize their DPCA method to nonlinear tasks by combining a neural autoencoder. They also discussed their training and inference methods to avoid re-training neural networks for different bandwidths. In section 5, the experiments showed performance of proposed framework in three tasks. All experiments assumed two data sources and compared the framework with three other baseline methods. The results showed that task-aware NDPCA performed similar to or better than all baselines in three tasks and had a graceful tradeoff between performance and bandwidth.

**Strengths:**

1.The idea of compressing data from distributed sources into different sizes according to their importance is novel and effective. The proposed formulation of the problem and its solution in linear setting serve as a good insight for nonlinear version of the problem.

2.The proposed training and inference methods of NDPCA do not require retraining for different bandwidth, which is convenient and saves computing and storage resources.

3.The experiments in this paper compare the proposed framework to three baseline methods in three different tasks. The authors comprehensively analyze the results to show the advantages of task-aware NDPCA in different practical situations.

**Weaknesses:**

1.The theoretical analysis in section 2 and 3 only covers situations where encoder, decoder and tasks are linear, which is not a strong support for NDPCA that works for non-linear situations.

2.The experiments assume only two data sources, which is a heavy restriction. The performance of NDPCA when there are more than 2 (or a very large number of) data sources are not presented or analyzed.

3.As described in the last paragraph of section 5, the autoencoders are poor at generalizing out-of-distribution data, which is also the weakness of NDPCA. This paper does not offer a solution to this problem, which will possibly limit the usefulness of NDPCA in practical scenarios where fresh data is generated and transmitted in real-time.

**Questions:**

1.What is the performance of NDPCA when there are more than 2 data sources?

2.What is the performance of NDPCA when using testing dataset? If it is too low, is there an insight why or any possible solutions?

3.In section 5, it is very interesting that NDPCA outperforms JAE, but the reason is only simply explained in the paper. Could you please elaborate a bit?

4.Could you provide a simple theoretical analysis or insights explaining why difference of DPCA reconstruction losses does not harms the result of non-linear tasks?

**Limitations:**

The authors have discussed the limitations.

---

> ### Author Rebuttal · Authors · 2023-08-09
>
> Thank you for the comments.
>
> Weakness:
>
> 1. The key focus of our study is to **harmonically combine linear DPCA modules and neural networks**, leveraging their different purposes. The linear DPCA module is designated to measure the importance of sources with singular values, and the neural networks are designed to process complicated real-world data and work harmonically with the DPCA modules. While there are no theoretical guarantees for their combination, we tested NDPCA on real-world datasets to demonstrate its capability.
>
> 2. (also question 1) Both our DPCA and NDPCA frameworks have been designed to accommodate multiple sources effectively. In particular, our DPCA linear formulation demonstrates its effectiveness with multiple sources, as outlined in lines 137-139 of the original submission. **We also add one additional experiment with NDPCA compressing 4 sources with different sizes of views**. Again, its performance was compared to that of a task-aware vanilla distributed autoencoder (DAE) in the same setting. The results, presented in the global review's attached PDF, reveal NDPCA's superiority over DAE in the Airbus dataset, and NDPCA with two sources slightly outperformed the one with four sources. The distribution of bandwidth to sources is unequal because each source has different importance to the task, which is related to the size of the view.
>
> 3. (and also question 2) **The results presented in the paper are based on testing sets**, emphasizing the case where fresh data is generated and transmitted in real time. We used data augmentation (using the albumentation package as described in the appendix) to aid the neural networks in adapting to unseen data better, bridging the gap of generalization.
>
> Questions:
>
> 3. We find the results interesting as well. Two perspectives explain the performance difference between NDPCA and JAE. First, NDPCA has two encoders, making its model parameters slightly larger than JAE, which may contribute to better performance. Second, NDPCA uses DPCA modules to compress representations Z, which can effectively remove noise injected in the data, giving it an advantage over JAE in noise removal and performance.
>
> 4. The reason is that after DPCA reconstruction, NDPCA uses a decoder to decode the data back to its original space. During training, the encoders and decoder are exposed to the DPCA reconstruction, while the other methods we tried in the appendix section do not, so the DPCA reconstruction only slightly harms the result of non-linear tasks. Of course, it is impossible to not harm the result of non-linear tasks as data is always compressed.

---

> > ### Comment · Reviewer_84yk · 2023-08-13
> > **Thanks for the response.**
> >
> > Thanks for your clarification on the problems! The answers have provided some insights on technical details of the paper, and the additional experiment on NDPCA compressing data from 4 sources. Although its still theoretically unclear why neural networks play an important part in NDPCA and the experiments have room for improvement, it could be a solid technique in applications.  I would increase my score.

---

> > > ### Author Response · Authors · 2023-08-21
> > > **Can you double-check the score?**
> > >
> > > A kind reminder to double-check whether the score is correctly saved.

---

### Official Review · Reviewer_uwj4 · 2023-07-07

**Soundness:** 2 fair
**Presentation:** 2 fair
**Contribution:** 1 poor
**Rating:** 3
**Confidence:** 4

**Summary:**

This paper studies compression in a distributed computing setting, named neural distributed principal component analysis(NDPCA). The proposed NDPCA can adapt to available bandwidth and flexibly allocates bandwidth to multiple sources according to their contribution to the final task. Experiments demonstrate the effectiveness of NDPCA on bandwidth allocation.

**Strengths:**

By dynamically distributing bandwidth among sensors, NDPCA implements a graceful trade-off between performance and bandwidth, enabling adaptive resource allocation. The experiments conducted in the paper demonstrate that NDPCA significantly improves the success rate of multi-view robotic arm manipulation by 9% and the accuracy of object detection tasks on satellite imagery by 14% compared to an autoencoder with uniform bandwidth allocation.

**Weaknesses:**

The proposed method is mainly a control algorithm that can adapt to multiple streams of data, which sound like a resource allocation algorithm. The neural processing of data is not essential here.

The experiments may be too simplistic to be convincing. For example, the three experiments are all in a two-view setting and may not provide a realistic assessment of bandwidth allocation capabilities. Besides, there is no comparison with prior works.

**Questions:**

In lines 198-199, “We observed that the autoencoder 199 automatically learns representations with small correlation”. Does this observation still exist when there are more encoders (>2)?

**Limitations:**

No particular limitations.

---

> ### Author Rebuttal · Authors · 2023-08-09
>
> Thank you for the feedback.
>
> Yes, our NDPCA can be interpreted as a resource allocation algorithm, but the key focus of our study is to **harmonically combine linear DPCA modules and neural networks**, leveraging their different purposes. The linear DPCA module measures the importance of sources using singular values, while the neural networks process complex, high-dimensional real-world data, making them essential for real-world applications.
>
> As stated in the related work section, to the best of our knowledge, our work is the only one focusing on **designing a framework to compress multi-sourced data to different compression levels using the same model.** Other previous works have centered on designing new neural architectures for multi-view image compression, using different neural network layers. Given this distinction in focus and setting, we chose not to compare NDPCA with other works and only compared it with JAE and DAE. **We also add one additional experiment with NDPCA compressing 4 sources with different sizes of views.** Again, its performance was compared to that of a task-aware vanilla distributed autoencoder (DAE) in the same setting. The results, presented in the global review's attached PDF, reveal NDPCA's superiority over DAE in the Airbus dataset, and NDPCA with two sources slightly outperformed the one with four sources. The distribution of bandwidth to sources is unequal because each source has different importance to the task, which is related to the size of the view.
>
> Questions:
>
> Yes, similar trends of small correlations also exist in our additional experiments with 4 sources

---

### Official Review · Reviewer_dRdb · 2023-07-09

**Soundness:** 2 fair
**Presentation:** 3 good
**Contribution:** 2 fair
**Rating:** 5
**Confidence:** 2

**Summary:**

This work targets compressing the correlated data to be communicated in a multi-sensor network. The multi-sensor network pipeline is defined as the following steps: (1) each edge sensor compresses the data and transmits it to a central node, and (2) the central node decompresses the data and passes it to a machine learning task for the final output. Specifically, the authors first formulate a task-aware distributed source coding problem based on the target application. Then, they provide a theoretical justification for the formulated problem and propose a task-aware distributed source coding framework. The experiments on CIFAR-10 denoising, multi-view robotic arm manipulation, and satellite image object detection validate that the proposed framework can achieve better accuracy as compared to an autoencoder baseline under the same compression ratio.

====Post Rebuttal===
Thanks for the rebuttal, I have read the author’s rebuttal. The rebuttal addressed my concern on "Scalability to different scenarios" by adding more explanation on Locate and Lift tasks. For the other two concerns (e.g., "Need more motiving applications" and "Lack of compression with recent neural compression methods for images"), I was not fully convinced by the rebuttal.

**Strengths:**

> + Clear problem formulation: the problem formulation with visual-friendly figures can help the readers easily understand the target application,
> + Providing theoretical analysis and algorithm table: the provided theoretical analysis and algorithm table helps the readers understand the proposed method in a more structured way.
> + Comprehensive analysis of experiment results: the experiment settings are well described, and comprehensive analyses are conducted to better show figures and tables.

**Weaknesses:**

> + Need more motiving applications: although pure theory work can also have a huge impact, it would be great to add more real-world motiving applications for justly that compressing data for multi-sensor networks is an important question, and the proposed method is the key enabler. For example, with more and more convenient Internet access, will the varying and limited bandwidth still be a problem for existing edge devices? Why assume the data will not be decoded for humans but only for computers? What is the key metric to benchmark the multi-sensor network can achieve decent performance? > 99% accuracy on some specific applications or do the machine learning task in a real-time manner? Why is the improvement by the proposed method meaningful? Will the whole pipeline with the proposed method achieve some specific common-agreed metric (e.g., 30FPS  real-time), and the pipeline without the proposed method cannot?
> + Lack of compression with recent neural compression methods for images: although this work does not only target compression for images, but at least two of the three benchmark datasets are pure image datasets. Since there exist some works that target encoding images into neural networks (e.g., https://arxiv.org/abs/2006.09661), it will be great to add the comparison and discussion with those works.
> + Scalability to different scenarios: In the task of “Locate and Lift”, the trends of different methods are not the same trend as the trends on the other two datasets. Specifically, the proposed NDPCA does not always achieve a higher success rate as compared to the baseline DAE. Is it caused by the task type being quite different from image denoising and object detection? Thus, the scalability of different scenarios is unclear.

**Questions:**

> + What is the cost of training the proposed NDPCA as compared to methods without training?

**Limitations:**

See "Weaknesses"

---

> ### Author Rebuttal · Authors · 2023-08-09
>
> Weakness:
> 1. In our study, we consider a scenario where satellites send data to a mission control center on Earth, using independent encoders due to their distance apart, with the control center having a joint decoder. Similarly, as the IoT era nears, factories will use sensors in distant locations, demanding efficient machine-learning models to process compressed data from these sensors jointly.
> As shown in Fig. 4 and the appendix, **our method can trade off human perception and computer features with a weighted loss, but in our main focus, we show results that are fully task-aware.**
> Our primary emphasis stems from the limited communication bandwidth. **To effectively execute machine learning tasks, the bandwidth should prioritize transmitting task-aware features.**
>  The key metric in the multi-sensor network setting is the performance of the task with limited bandwidth as we argue that bandwidth is the bottleneck of such settings. In other settings, real-time inference may be essential, but it is not the focus of the paper. The NDPCA framework only adds negligible computation which we elaborate on later.
>
> 2. As stated in the related work section, to the best of our knowledge, our work is the only one **focusing on designing a framework to compress data to different compression levels by the same models.** Other previous works have centered on designing new neural architectures for multi-view image compression, using different neural network layers. Given this distinction in focus and setting, we chose not to compare NDOCA with other works and only compared it with JAE and DAE.
>
> 3. The Locate and Lift experiment is a reinforcement learning behavior cloning task that is **highly influenced by the initial environmental conditions during both the training and testing phases**. This is also why we showcase the performance of runs from multiple random seeds. The sensitivity is evident from the substantial variance in performance observed in various RL papers, such as the soft-actor-critic paper by Haarnoja et al. in 2018 (Figure 1). In the realm of RL research, researchers typically compare the average performance of different methods across multiple random seeds. In this paper, **the NDPCA method consistently outperforms DAE on average in our specific experimental setting.**
>
> Questions:
>
> NDPCA has additional DPCA modules compared to vanilla DAE. The DPCA module performs singular value decomposition (SVD) on a batch of data. The memory and computational time overhead of using SVD during training are neglectable compared to the backpropagation of deep neural networks. The reason is that SVD is only a series of matrix operations and the dimension of matrices in SVD is “batch size x dimension of representations”, which makes it easy to compute.

---

### Official Review · Reviewer_A8qh · 2023-07-12

**Soundness:** 2 fair
**Presentation:** 3 good
**Contribution:** 2 fair
**Rating:** 4
**Confidence:** 3

**Summary:**

This paper proposes a solution for multi-view machine learning with distributed computing and limited bandwidth. Different views of data are encoded and then compressed to be transmitted to the decoder for learning tasks. It assigns higher bandwidth for data with better quality to make a tradeoff between different quality of data. Experiment results show higher psnr or reconstruction rate or accuracy than prior art.

**Strengths:**

The problem this paper tries to solve is important. The multi-view distributed learning will improve over single-view learning and the communication can be a bottleneck for the system.



**Weaknesses:**

The experimented model and data is somehow outdated. For example, CIFAR10 is an old dataset and the paper does not provide the details on the classification network or the accuracy for classification. Detection dataset (airbus) and model (YoloV1) should be updated to larger and modern ones, such as COCO and YoloV6, V7or V8. Though interesting, the method seems, at least to me, to be working on toy examples and models.

**Questions:**

Why for CIFAR10, only the reconstruction PSRN is reported, not the accuracy of the classification? If this is the case, we could use any image dataset, not necessarily classification dataset, such as DIV2K for image reconstruction.

---

> ### Author Rebuttal · Authors · 2023-08-09
>
> We thank the reviewer for their insightful feedback.
>
> Weakness:
>
> Regarding the use of CIFAR10, we acknowledge its age but included it as a toy example for quick iteration and sanity checks due to its manageable size. Thus, we can try different methods to improve uncorrelatedness and linear compressibility, as presented in the ablation study in the appendix.
>
> Regarding the airbus detection with Yolov1, we also experimented with Yolov8 but found no significant difference between the two. Additionally, we opted for Yolov1 due to its transparency and ease of modification for our setting, supported by abundant open-source resources available online. It is important to note that our paper primarily focuses on **the methodology of data compression for multiple sources**, rather than showcasing state-of-the-art computer vision models.
>
> Questions:
>
> In short, CIFAR10 PSNR was selected for its convenience in quick iteration and sanity checks during our research. We use CIFAR-10 as a toy example to demonstrate the use of NDPCA in the presence of sources with unequal importance to the task. Due to the **simplistic nature of the classification task, which only requires 4 bits (digit 0-9) as the information bottleneck**, we choose reconstruction as our “task”, making it more suitable to showcase the performance across a range of available bandwidth. Moreover, we utilized image reconstruction to showcase the ability of NDPCA in task-agnostic settings. While choosing other image datasets is feasible, we already demonstrated the results of NDPCA in the computer vision domain with our Airbus experiments.

---

### Official Review · Reviewer_SHcb · 2023-07-15

**Soundness:** 3 good
**Presentation:** 2 fair
**Contribution:** 3 good
**Rating:** 5
**Confidence:** 2

**Summary:**

This paper presents the neural distributed principal component analysis (NDPCA) method that compresses features from multiple sensor sources with a given total bandwidth limit. NDPCA carries the following novelties. First, it is task-aware. The algorithm trains the compression networks by minimizing the final task. Second, the single model can compress the features at different bandwidth limits. This is achieved by training the networks at the maximum available bandwidth and picking the largest components based on the available bandwidth at inference time.

The authors evaluated the NDPCA method on three different tasks: 1) denoising of CIFAR-10 images, 2) multi-view robotic arm manipulation, and 3) object detection from satellite imagery. They compared NDPCA against the following baselines: 1) vanilla distributed PCA with equal bandwidth allocation, 2) joint PCA that jointly encodes all sources, and 3) task-agnostic NDPCA. The evaluation result shows NDPCA achieves much better task performance than vanilla distributed PCA and can even outperform joint PCA.

**Strengths:**

* This paper presents a sound method to compress features from multiple sources given limited bandwidth. The method outperforms the baselines by doing the compression in a task-aware manner.

* The evaluation used three real-world applications.

**Weaknesses:**

* I have some doubts about the problem setup used in the paper. The paper assumes there is a total availabe bandwith that is shared by all sensor sources, and there is no per-source bandwidth limit (i.e., one can allocate all the bandwith to one single source). It will be useful to provide some references to support this assumption. My concern with this setup is that, to have the communication bottleneck shared by all sources, those are likely very close to each other. If the sensors are close to each other, one can build a joint encoder that encodes the features from all sources jointly, which makes joint PCA a realistic option.

* The evaluation contains only setups with two sources. It will be useful to provide results showing the method's performance with more sources.

**Questions:**

It will be useful to provide some references to support the assumption of bandwidth limit.

**Limitations:**

The authors have adequately addressed the limitations.

---

> ### Author Rebuttal · Authors · 2023-08-09
>
> We express our gratitude to the reviewer for their valuable feedback, and we will address each point in detail.
>
> Weakness:
>
> 1-1. Our study was inspired by the scenario presented in the introduction, involving multiple distant satellites transmitting data to an Earth-based mission control center. Employing independent encoders for the satellites and a joint decoder for the control center, we mimicked this setup. Given the upcoming IoT era where scattered sensors gather data, our research focused on a similar challenge: effectively utilizing compressed data from individual sensors for joint decoding and processing, which holds significance for future applications.
>
> 1-2. Regarding the reviewer's concern about the per-source bandwidth limit, **we would like to highlight that our proposed framework, NDPCA, is adaptable to this scenario.** NDPCA is capable of measuring the importance, represented by singular values, of each dimension from all sources. As a result, we can allocate bandwidth to each source effectively using greedy allocating algorithms, prioritizing the most important available source per source constraints.
>
> 2. Both our DPCA and NDPCA frameworks have been designed to accommodate multiple sources effectively. In particular, our DPCA linear formulation demonstrates its effectiveness with multiple sources, as outlined in lines 137-139 of the original submission. **We also add one additional experiment with NDPCA compressing 4 sources with different sizes of views.** Again, its performance was compared to that of a task-aware vanilla distributed autoencoder (DAE) in the same setting. The results, presented in the global review's attached PDF, reveal NDPCA's superiority over DAE in the Airbus dataset, and NDPCA with two sources slightly outperformed the one with four sources. The distribution of bandwidth to sources is unequal because each source has different importance to the task, which is related to the size of the view.
>
> Question:
>
> We examine a scenario where there's an uplink bandwidth limitation, essentially a collective constraint shared by all sensors. For instance, in an IoT setup with 100 sensors and a single central node, this constraint translates to a reception bandwidth limitation for the central node, less than or equal to the combined bandwidth of all 100 nodes. Such a situation is typical in wireless sensor networks.
>
> Reference: P. Liu *et al.*, "Training Time Minimization in Quantized Federated Edge Learning under Bandwidth Constraint," *2022 IEEE Wireless Communications and Networking Conference (WCNC)*.
>
> Furthermore, in the satellite network scenario described in our introduction section and our object detection experiments, the bandwidth available to satellites is not on par with that of 5G wireless networks. Satellite network providers like Starlink offer internet speeds ranging from 50 to 250 Mbps, whereas 5G networks can reach speeds of up to Gbps.

---

> > ### Comment · Reviewer_SHcb · 2023-08-17
> > **Thank you for your response**
> >
> > Thank you for your response. I will keep my score.

---

### Official Review · Reviewer_V3iA · 2023-07-26

**Soundness:** 3 good
**Presentation:** 4 excellent
**Contribution:** 3 good
**Rating:** 6
**Confidence:** 3

**Summary:**

This paper proposes a distributed task-specific compression method called NDPCA, composed of both a neural network autoencoder and a linear PCA reconstruction. Given multiple sources of data, NDPCA first compresses the information separately using different independent neural network encoders. Next, it applies a linear distributed encoder based on PCA, which further bottlenecks the information. For each neural encoded source, it projects it into a PCA subspace, such that the total number of dimensions used is equal to a predefined bandwidth $m$. Each source is allocated a different number of dimensions, based on the ranking in the top $m$ singular values. The information is decoded twice, first reprojected back from the PCA subspaces and secondly using a neural network decoder that goes back to the original sources space. The reconstructed sources are input to a task specific network. The authors propose to learn the autoencoders in NDPCA using task-aware losses, that is the task loss should be the same with the reconstructed sources as with the original sources. One of the benefits of the proposed approach is that, because of the use of PCA as an intermediate reconstruction step, one can dynamically choose the total number of eigenvectors used (and, consequently, the bandwidth) without a need for retraining the model. The authors provide a theoretical analysis of their approach and multiple experiments on a number of different tasks that support the methodological choices.

**Strengths:**

1. The paper is written clearly, well structured and easy to follow.
2. The proposed method, while simple, is novel and achieves good results. The random DPCA module is a nice idea.
3. The experimental evaluation is done on multiple different tasks, which demonstrated the applicability of the proposed framework.
4. I like the negative results discussion provided in the paper, relating to (1) uncorrelatedness and (2) linear compressibility.


**Weaknesses:**

1. While the framework presented is described to work for any number of input sources, all the experiments are conducted by considering only 2. I would have loved to see how the method behaves for a larger number of sources.

**Questions:**

1. By using SVD in training, how much overhead does it introduce? In terms of memory/time vs when not using random DPCA in training.
2. How important is the batch size? You state in the limitations that SVD can become unstable for small batch sizes, but do the results improve with a higher batch size?

**Limitations:**

The limitations are addressed by the authors.

---

> ### Author Rebuttal · Authors · 2023-08-09
>
> We thank the reviewer for appreciating our paper.
>
> Weakness:
>
> Both our DPCA and NDPCA frameworks have been designed to accommodate multiple sources effectively. In particular, our DPCA linear formulation demonstrates its effectiveness with multiple sources, as outlined in lines 137-139 of the original submission.
> **We also add one additional experiment with NDPCA compressing 4 sources with different sizes of views.** Again, its performance was compared to that of a task-aware vanilla distributed autoencoder (DAE) in the same setting. The results, presented in the global review's attached PDF, reveal NDPCA's superiority over DAE in the Airbus dataset, and NDPCA with two sources slightly outperformed the one with four sources. The distribution of bandwidth to sources is unequal because each source has different importance to the task, which is related to the size of the view.
>
> Questions:
> 1. The memory and computational time overhead of using SVD during training is neglectable compared to the backpropagation of deep neural networks. The reason is that SVD is only a series of matrix operations and the dimension of matrices in SVD is “batch size x dimension of representations”, which makes it easy to compute.
> 2. Yes, a small batch size may cause the SVD to be ill-conditioned and unstable as the matrix might have multiple 0 singular values. For a higher batch size, **we do not observe any significant improvement compared to a sufficiently large (stable) batch size**.
> a

---

> > ### Comment · Reviewer_V3iA · 2023-08-19
> >
> > After carefully reading the other reviews and the authors' rebuttal, I decided to maintain my initial rating. I think this paper has merits to be accepted and I do not agree with the sentiment of other reviewers that were, in my opinion, too dismissive of the paper.

---

### Author Rebuttal · Authors · 2023-08-09

This is the attached file to show the latest results from our additional experiments with 4 data sources.

---

### Decision · Program_Chairs · 2023-09-21

**Decision:**

Accept (poster)

**Comment:**

The paper proposes a neural distributed principal component analysis (NDPCA). It's a distributed compression framework composed of independent encoders and a joint decoder. NDPCA can compress data from multiple sources to any available bandwidth. Reviewers were generally positive about the submission. The concern raised by the more concerned reviewer were according to the opinion of the AC addressed in the author feedback.